# Structural Biology-Based Exploration of Subtype-Selective Agonists for Peroxisome Proliferator-Activated Receptors

**DOI:** 10.3390/ijms22179223

**Published:** 2021-08-26

**Authors:** Hiroyuki Miyachi

**Affiliations:** Lead Exploration Unit, Drug Discovery Initiative, The University of Tokyo, 7-3-1 Hongo, Bunkyo-ku, Tokyo 113-0033, Japan; miyachi_hiroyuki@mol.f.u-tokyo.ac.jp

**Keywords:** peroxisome proliferator-activated receptor, PPAR agonist, structural biology, ligand superfamily concept, helix 12 holding induction concept

## Abstract

Progress in understanding peroxisome proliferator-activated receptor (PPAR) subtypes as nuclear receptors that have pleiotropic effects on biological responses has enabled the exploration of new subtype-selective PPAR ligands. Such ligands are useful chemical biology/pharmacological tools to investigate the functions of PPARs and are also candidate drugs for the treatment of PPAR-mediated diseases, such as metabolic syndrome, inflammation and cancer. This review summarizes our medicinal chemistry research of more than 20 years on the design, synthesis, and pharmacological evaluation of subtype-selective PPAR agonists, which has been based on two working hypotheses, the ligand superfamily concept and the helix 12 (H12) holding induction concept. X-ray crystallographic analyses of our agonists complexed with each PPAR subtype validate our working hypotheses.

## 1. Nuclear Receptors

Nuclear receptors (NRs) are ligand-dependent transcription factors that modulate diverse aspects of development, reproduction, and energy homeostasis. This receptor superfamily includes receptors for vitamin D, steroid hormones, thyroid hormones and retinoids, as well as a large number of orphan receptors. NRs are composed of six functionally distinct regions (termed A to F) (Figure 1A). The *N*-terminal AB region is highly variable and contains a constitutionally active transactivation function-1 (AF-1) motif. The central C region (a DNA-binding region) is highly conserved among NRs and contains two zinc finger motifs that make contact with specific nucleotide sequences, termed hormone response elements. The C-terminal D, E and F regions are required for ligand binding and receptor dimerization. In most NRs, these regions also contain a second highly conserved transcriptional activation function-2 (AF-2) motif, which is important for ligand-dependent transcription.

In the basal state, NRs are functionally inactive because they are tightly bound with corepressors, such as NCoR and SMRT. Upon NR binding with an activating ligand, the corepressor dissociates and then coactivators, such as SRC-1 and NCoA, are recruited and the AF-2 helix located in the F region is stabilized to initiate transcription (Figure 1B) [1]. Over the past three decades, much attention has been focused on a subgroup of NRs, the peroxisome proliferator-activated receptors (PPARs). 

## 2. Peroxisome Proliferator-Activated Receptors

PPARs are activated by endogenous unsaturated and saturated fatty acids and by synthetic ligands [2]. There are three PPAR subtypes: PPARα, PPARδ, and PPARγ. Each PPAR subtype is expressed in a tissue-specific manner. PPARα is mostly expressed in tissues involved in lipid oxidation, such as liver and kidney. PPARγ is expressed in adipose tissue, macrophages and vascular smooth muscle, and also in tumors originating from various organs. PPAR*δ* is expressed in adipose tissue, skeletal muscle, heart, etc. [3]. 

Upon ligand binding, PPARs heterodimerize with another nuclear receptor, retinoid X receptor (RXR), in the nucleus, and the heterodimers regulate genes expression, such as carnitine palmitoyl acyl-CoA transferase 1A (*CPT1A*), angiopoietin-like protein 4 (*ANGPTL4*) and adipocyte differentiation-related protein (*ADRP*) by binding to specific consensus DNA sequences, termed peroxisome proliferator responsive elements (PPREs) in the promoter regions of target genes. The structural basis of PPREs is a direct repeat of the hexameric AGGTCA recognition motif, separated by one nucleotide (termed DR1) [4].

## 3. Pleiotropic Effect of PPARs

PPAR subtypes play a key role in lipid, lipoprotein and glucose homeostasis. PPARα regulates genes involved in fatty acid uptake, β-oxidation, and ω-oxidation. It downregulates apolipoprotein C-III, which regulates triglyceride hydrolysis by lipoprotein lipase, and also regulates genes involved in reverse cholesterol transport, such as apolipoprotein A-I and apolipoprotein A-II [5]. PPARδ activation regulates HDL cholesterol levels, and it influences glycemic control [6,7,8]. PPARδ activation markedly improves glucose tolerance and insulin resistance [9]. PPARγ is a master regulator of adipocyte differentiation, but recent molecular studies have indicated that its activation is also linked to the expression of many important genes that affect energy metabolism, such as TNF-α, leptin, and adiponectin [10]. PPARγ also promotes cell cycle arrest by inhibiting cyclin-dependent kinase activity in several tumor cell lines [11].

Recent extensive biological studies clearly disclosed that PPARs function beyond metabolism. Each PPAR subtype plays major roles in a broad spectrum of biological processes, including cell proliferation and differentiation, fatty acid and eicosanoid signaling, bone formation, tissue repair and remodeling, insulin sensitivity [12].

The above examples demonstrate that PPARs are important pleiotropic NRs and attractive molecular targets for the treatment of various diseases. Therefore, we think it is important to develop potent and PPAR subtype-selective ligands as tools to investigate the detailed functions of individual PPARs. In this review, we summarize in historical order our structural development studies to create PPAR subtype-selective agonists. The discovery of many types of PPAR ligands indicates the validity of our strategy to create subtype-selective NR agonists.

## 4. Working Hypothesis of the NR Ligand Superfamily

For over twenty years, we have been engaged in NR ligand structure research, which has been based on our working hypothesis of the NR ligand superfamily [13]. The structural and functional features of the many different NRs are similar; therefore, we speculate that all NRs are derived from a single ancestral protein and have structurally evolved to fit various kinds of endogenous NR ligand. Similar evolution of NR ligands would have occurred from an ancestral ligand to form a superfamily of NR ligands, even though they now have diverse structures and functions. Based on this hypothesis, NR ligand structures are divided into two types, basic framework and branch. A common hydrophobic skeleton or a branch that fits into the ligand binding pocket of the basic ancestral protein are characteristic structural motifs that provide NR selectivity. PPAR subtype-selective agonists have certain unique structures associated with subtype selectivity. Examples include, thiazolidine-2,4-dione (TZD) and related structures, exemplified by pioglitazone, for PPARγ [Figure 2 (1)], the 2,2-dialkyl(usually dimethyl)phenoxyacetic acid structure, exemplified by fenofibrate, for PPARα [Figure 2 (2)], and the 2,2-unsubstituted phenoxyacetic acid structure, exemplified by GW-501516, for PPARδ [Figure 2 (3)]. However, based on our working hypothesis, we predict that various kinds of subtype-selective, dual-, and pan-agonists can be created by starting with a common chemical framework as a template.

Based on our hypothesis, we have successfully created various kinds of subtype-selective PPAR agonist, including KCL (PPAR α-selective agonist) [14], APHM-19 (PPAR α-selective agonist) [15], TIPP-401 (PPAR α/δ dual agonist) [16], APHM-13 (fluorescent PPAR α/δ dual agonist) [17], TIPP-204 (PPAR δ-selective agonist) [18], TIPP-703 (PPAR pan agonist) [19], MO-4R (PPAR γ-selective agonist) [20], MEKT-21 (PPAR γ-selective partial agonist) [21], and MEKT-75 (PPAR γ-selective partial agonist) [22]. In addition, in collaboration with Dr. Oyama (Yamanashi University, Japan) and Dr. Shimizu (Tokyo University, Japan), we have solved many X-ray crystallographic structures of PPAR subtypes complexed with our ligands [23,24,25]. These structural biology studies have been integral to our medicinal chemistry research (Figure 3).

## 5. Synthesis of Our Ligands

We have designed and synthesized a series of phenylpropanoic acids to develop structurally new PPAR subtype-selective agonists. The agonist synthetic scheme is depicted in Scheme 1. Benzylcarbamoyl-tethering compounds, such as KCL (Figure 2), were prepared by synthesis route I. 5-Formylsalicylic acid (4) was esterified and alkylated to obtain compound 5. The formyl group of 5 was reduced with sodium borohydride (6), then bromination of the hydroxyl group afforded the bromomethyl derivative (7). Evans asymmetric alkylation [26] was performed, followed by hydrogenolysis to afford the key intermediate 9. This was amidated (10), and the chiral auxiliary removed to afford the desired (*S*)-configuration product. The antipodal (*R*) enantiomer was prepared via similar procedures, using (*S*)-*N*-butyryl-4-benzyloxazolidinone as the chiral reagent. Benzamidomethyl-tethering compounds, such as TIPP-401 (Figure 2), were synthesized by synthesis route II. The carboxyl group of key synthetic intermediate 9 was reduced with borane-THF complex, followed by oxidation with activated manganese dioxide to afford the formyl derivative 11. This was amide-alkylated with benzamide derivatives [27], followed by removal of the chiral auxiliary to afford the desired (*S*)-configuration product. The (*R*) enantiomer was also prepared via similar procedures, using (*S*)-*N*-butyryl-4-benzyloxazolidinone as the chiral reagent.

Reagents and conditions.

(Synthesis route I) (a) (1) BnBr, KHCO_3_, DMF, rt. (2) R^2^I, K_2_CO_3_, DMF, rt. (b) NaBH_4_, EtOH, rt. (c) PBr_3_, ether, 0 °C. (e) (1) (*R*)-4-benzyl-3-butyryloxazolidin-2-one, LiHMDS, THF, −40–10 °C, (2) benzyl 5-bromomethyl-2-methoxybenzoate, THF, −40 °C–−10 °C. (e) H_2_, 10% Pd-C, AcOEt, rt. (f) (1) ethyl chloroformate, TEA, THF, −10 °C, (2) R^1^-benzylamine, THF, −10 °C–rt. (g) LiOH H_2_O, H_2_O_2_, THF-H_2_O, 0 °C.

(Synthesis route II) (a) (1) BnBr, KHCO_3_, DMF, rt. (2) R_2_I, K_2_CO_3_, DMF, rt. (b) NaBH_4_, EtOH, rt. (c) PBr_3_, ether, 0 °C. (e) (1) (R)-4-benzyl-3-butyryloxazolidin-2-one, LiHMDS, THF, −40–10 °C, (2) benzyl 5-bromomethyl-2-methoxybenzoate, THF, −40–10 °C. (e) H_2_, 10% Pd-C, AcOEt, rt. (h) (1) BH3-tetrahydrofuran, THF, 0 °C. (2) activated MnO_2_, CH2Cl2, rt. (i) 2-fluoro-4-trifluoromethylbenzamide, triethylsilane, trifluoroacetic acid, toluene, reflux. (g) LiOH H_2_O, H_2_O_2_, THF-H_2_O, 0 °C.

## 6. PPARα-Selective Agonist: From KCL to APHM-19

In 1996, we started our PPAR ligand discovery program. We aimed to develop PPARα-selective agonists as anti-dyslipidemic agents by designing and synthesizing a series of substituted phenylpropanoic acid derivatives using as lead compound, KRP-297, a unique TZD derivative with PPARγ/α dual agonist activity [28,29,30]. KRP-297 was the first dual active thiazolidinedione (also known as glitazone), a TZD class insulin sensitizer, that binds directly to and activates not only PPARγ, but also PPARα with almost equal affinity. Selective PPARγ activation is a characteristic feature of classical glitazones, including troglitazone, pioglitazone (**1**), and rosiglitazone. We anticipated that replacement of the thiazolidine-2,4-dione ring of KRP-297 with another acidic functionality, such as carboxylic acid, might favor PPARα selectivity. KRP-297 can be divided into three structural regions: (A) the acidic head, (B) the tether region, and (C) the hydrophobic tail. We performed chemical modification of (A) and (B) and obtained KCL, (*S*)-2-[4-methoxy-3-(4-trifluoromethylbenzylcarbamoyl)phenylmethyl]butyric acid [14,31,32,33] as a potent and PPARα-selective agonist (Figure 4B).



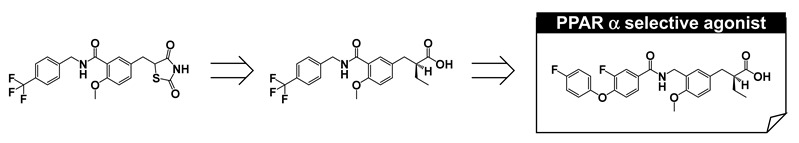



To our surprise, KCL possesses high species-selectivity for transactivation of human PPARα. KCL activates human, dog, and rat PPARα with EC_50_ values of 0.06, 0.16, and 5.2 μM, respectively. Mutation studies defined specific interaction between the isoleucine at position 272 (Ile272), which is located in the H3 region of the human PPARα ligand binding domain (LBD), and the hydrophobic tail of KCL [34]. The corresponding amino acid residue in the rat PPARα LBD is the sterically more bulky phenylalanine. Clinical trials of a KCL-related compound proceeded to phase IIa but were then discontinued.

To further strengthen PPARα agonist activity, we performed a computational docking study to construct binding models of TIPP-401 (a PPARα/δ dual agonist described later) complexed with the full-length human PPARα LBD or the human PPARδ LBD. We noticed that the shapes of the hydrophobic pockets hosting the hydrophobic tail of TIPP-401 (4-trifluoromethyl group) were somewhat different, i.e., the hydrophobic pocket of human PPARα is wider than that of human PPARδ (Figure 4A). Therefore, we predicted that if we could introduce a sufficiently bulky substituent instead of the 4-trifluoromethyl group, we would be able to create a more potent human PPARα-selective agonist, because the introduced substituent would enhance the interaction with the hydrophobic pocket of the human PPARα LBD. Based on this hypothesis, we discovered a more potent agonist, APHM-19, bearing a bulky 4-fluorophenoxy group in the hydrophobic tail (Figure 4B). We recently solved the X-ray crystallographic structure of APHM-19 complexed with the human PPARα LBD by the ligand-exchange soaking method [35]. This clearly indicated that the 4-fluorophenoxy group of APHM-19 docks in the hydrophobic pocket composed from Val332, Val255, Leu254, Glu251, Leu247, and Ile241. We also found that the proximal fluorine atom interacted hydrophobically with the side chain of Cys275 (Figure 5C).

To investigate the therapeutic potential of APHM-19, we evaluated its ability to block progression of nonalcoholic steato-hepatitis (NASH) in an animal model [36,37]. Although the pathogenesis of NASH has not been fully elucidated, the “two-hit” theory is widely accepted [38]. The first hit is the accumulation of fatty acids in the liver to cause steatosis. Steatosis causes relatively mild pathology; however, concomitant secondary cellular stress can progress steatosis to steatohepatitis [39,40]. The results are summarized in Figure 5A,B. Aspartate amino transferase (AST), alanine transaminase (ALT), and alkaline phosphatase (ALP) are well-known markers of liver function [41]. Their levels were significantly higher in the NASH rats compared with choline-deficient high-fat diet-fed rats. In the 1 mg/kg/day APHM-19-treated NASH group, liver function was improved compared with the NASH group, although the enzyme levels were still increased compared with normal chow-fed rats. These in vivo findings indicate that our PPARα-selective agonist can block, or at least delay, the progression of NASH. Therefore, PPARα-selective agonists have therapeutic potential for the treatment of NASH.

## 7. PARα/δ-Dual Agonist: TIPP-401

We next aimed to develop a PPARα/δ-dual agonist that would effectively activate both PPARα and PPARδ subtypes. Pharmacological evidence indicates that PPARα regulates the expression of genes involved in lipid and lipoprotein homeostasis, while PPARδ plays a key role in lipid metabolism and insulin resistance. We expected that a compound that effectively activates both PPARα- and PPARδ-subtypes might have additive and/or synergistic positive effect(s) in the treatment of metabolic syndrome. We speculated that small manipulations of the KCL structure would affect activities towards both PPARα and PPARδ subtypes. We reconsidered the structure–activity relationships of KCL derivatives and noted that a flexible tether has the tendency to decrease PPARα activation but also to activate PPARδ. We focused our attention on a hybrid type tether, i.e., –CO–NH–CH_2_–, and found that the compound with this tether increased both PPARα and PPARδ activity. Slight structural modification at the hydrophobic tail afforded the PPARα/δ dual agonist, TIPP-401 ((*S*)-2-{3-[(2-fluoro-4-trifluoromethyl-benzoylamino)methyl]-4-methoxybenzyl}butyric acid). A clear enantio-dependency of the transactivation activity of both PPAR subtypes was found. Like KCL, the (*S*) conformer exhibited potent transactivation activity towards both PPARα and PPARδ, while the antipodal (*R*) isomer was far less potent.



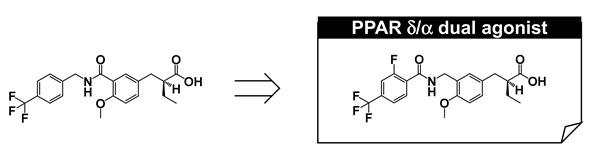



We succeeded in obtaining the X-ray crystallographic structure of TIPP-401 complexed with human PPARδ LBD. Hydrogen bonding was observed between the carbonyl oxygen of TIPP-401 and threonine 288 (T288) of PPARδ (Figure 6), while such hydrogen bonding was not found between the carbonyl oxygen of the KCL-type tether and T288. This might be a reason why the change of tether from amide type to reversed-amide type enhanced the PPARδ transactivation activity.

As described above, KCL shows species differences for PPARα. We evaluated the species-selectivity profile of PPARα transactivation by TIPP-401. As shown in Figure 7A, TIPP-401 activated human and mouse PPARα with EC_50_ values of 10 nM, and 1000 nM, respectively. Thus, the transactivation activity of TIPP-401 for PPARα was approximately 100-fold less in mice than in humans, demonstrating a species preference for humans. We, therefore, performed a mutagenesis study to evaluate the species-selectivity profile of TIPP-401. We found that Ile272 of human PPARα was also responsible for human-selective PPARα activation by TIPP-401 (Figure 7B).

Unfortunately, we have not succeeded in obtaining the X-ray crystallographic structure of TIPP-401 complexed with the human PPARα LBD. However, the crystal structure of APHM-19 complexed with the human PPARα LBD, as described above, clearly indicates its importance. The side chain alkyl group of Ile272 hydrophobically interacts with the phenoxyphenyl oxygen of APHM-19 (Figure 7C). Therefore, we speculated that the trifluoromethyl group of TIPP-401 also hydrophobically interacts with the side chain alkyl group of Ile272. In the mouse PPARα LBD, the corresponding amino acid is phenylalanine. The side chain benzyl group of phenylalanine might be too large to hydrophobically interact with TIPP-401.

TIPP-401 exhibits high nuclear receptor selectivity for PPARα and PPARδ because it did not significantly activate Vitamin receptor, PPARγ, LXRα, RARα or RXRα at concentrations up to 1 μM under the experimental conditions used (Figure 8A). To demonstrate the ability of TIPP-401 to activate genes that have PPREs in their promoter regions, we examined changes in expression of representative PPAR-regulated genes in human hepatocellular carcinoma Huh-7 cells. We chose carnitine palmitoyl acyl-CoA transferase 1A (*CPT1A*), angiopoietin-like protein 4 (*ANGPTL4*) and adipocyte differentiation-related protein (*ADRP*) as representative PPAR target genes. These human genes possess PPREs in their promoter regions [42,43]. We investigated the effects of fenofibrate (Feno; a PPARα-selective agonist), GW-501516 (GW; a PPARδ-selective agonist) and troglitazone (Tro; a PPARγ-selective agonist) on transactivation of these genes. The mRNA levels of these genes were augmented subtype selectively by treatment with 50 μM Feno, 0.1 μM GW, and 10 μM Tro (Figure 8B). These results indicated that each PPAR subtype is functionally active in Huh-7 cells. Treatment with 0.1 μM TIPP-401 augmented *CPT1A* gene expression to an extent comparable with that of obtained with 0.1 μM GW, and had little effect on *ANGPTL4* expression, probably because of the weak activity of TIPP-401 towards PPARγ. These results indicate that TIPP-401 is an effective PPARα/δ dual agonist at the cellular level.

## 8. PPARδ-Selective Agonist: TIPP-204

Our next focus was to create a PPARδ-selective agonist. Following the description of GW-501516 (3), several PPARδ-selective agonists have been reported, although most are derivatives of GW-501516, such as (2-methyl)phenoxyacetic acid derivatives [7,44,45,46,47,48]. We aimed to construct phenylpropanoic acid-type PPARδ-selective agonists, based on the PPARα/δ dual agonist, TIPP401.



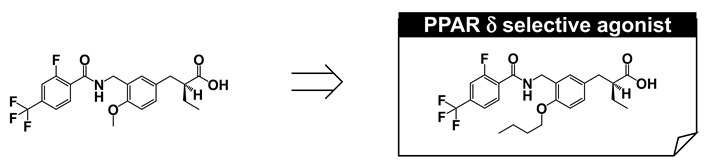



We considered the X-ray crystallographic analysis of PPARδ complexed with a natural unsaturated fatty acid, eicosapentaenoic acid (EPA) [49]. EPA binds to the cavity in two distinct conformations, i.e., “tail-up” and “tail-down” conformations. The carboxyl group and the first eight carbon units take almost the same configuration in both conformations. However, the distal hydrophobic tail of the tail-up EPA conformer is bent upwards into the upper cavity of the Y-shaped pocket, while in the tail-down conformer, the hydrophobic tail is bent downwards into the bottom cavity of the Y-shaped pocket. Our design was as follows. If we could connect one more sterically bulky hydrophobic side chain to the backbone of TIPP-401, directed towards the upper cavity of PPARδ, it should have the effect of strengthening PPARδ activity. We therefore prepared various 3-(4-alkoxyphenyl)propanoic acids, and generated the compound TIPP-204 ((*S*)-2-{3-[(2-fluoro-4-trifluoromethylbenzoylamino)methyl]-4-buthoxybenzyl}butyric acid). We obtained a clear structure–activity relationship confirming that the subtype-selectivity largely depended on the nature of the substituents, as expected (Figure 9A). TIPP-204 exhibited extremely potent PPARδ transactivation activity, comparable with or even superior to that of the known PPARδ-selective agonist, GW-501516.

To understand the significance of the *n*-butoxy group introduced at the alkoxy side chain of TIPP-204, we solved the X-ray crystallographic structure of TIPP-204 complexed with the human PPARδ LBD. We found that the *n*-butoxy group of TIPP-204 is positioned in the small cavity formed by Ile328, Leu303 and Val298. Val298 forms the bottom of the cavity, and Leu303 and Ile328 form the sides of the cavity (Figure 9B). A C4 *n*-butoxy group has the optimal length and further elongation decreases the activity.

Although X-ray crystallography was adequate to identify the key amino acids for human PPARδ (hPPARδ) activity, it was not appropriate to investigate the detailed contributions of the three human PPARδ amino acids to TIPP-204 binding. Reporter assays using a set of GAL4-fusion mutant human PPARs (Figure 10A) indicated that the potent transactivation of PPARδ by TIPP-204 was retained in the L303M mutant, decreased to some extent in the I328M mutant, but greatly decreased in the V298M mutant (approximately 30 times less potent) (Figure 10B). These three mutations all decrease the size of the binding cavity hosting the *n*-butoxy group of TIPP-204, and the rank order of the effect on PPARδ activity was Val298 > Ile328 > >Leu303. In contrast, for GW-501516, the mutant activities showed that the most important amino acid was different from that for TIPP-204. The transactivation of PPARδ by GW-501516 was retained in the V298M and L303M mutants, but greatly decreased in the I328M mutant (approximately 20 times less potent). Therefore, unlike TIPP-204, Val298 of PPARδ is less important than Ile328 for PPARδ selectivity of GW-501516. TIPP-401 showed the same tendency as GW-501516.

These mutation studies clearly indicated that the selectivity of human PPARδ for TIPP-204 is mainly attributed to favorable interactions between the *n*-butoxy group of TIPP-204 and Val298.

## 9. Fluorescent PPARα/δ Dual Agonist: APHM-13

We have made extensive use of functional assays in our PPAR agonist discovery research, i.e., transient transactivation assays with CMX-GAL4N-hPPAR LBDs as the recombinant reporter gene. However, affinity-based assays are important because they can determine the direct binding nature of the test compound. Therefore, we designed and synthesized a fluorescent human PPARα/δ dual-agonist termed APHM-13, which proved suitable for use in homogeneous fluorescence polarization assays [50]. The planar fluorophore pyrene was selected based on the X-ray crystallographic structure of the complex of our human PPARα/δ dual-agonist, TIPP-401, with the human PPARα LBD (Figure 10A,B), in which the distal hydrophobic 2-fluoro-4-trifluoromethylphenyl group of TIPP-401 is located in the cavity of the Y-shaped pocket of the human PPARα-LBD (Figure 11A). As expected, X-ray crystallography clearly showed that the pyrene ring of APHM-13 was tightly hosted in the hydrophobic cavity (Figure 11C,D). As predicted, the fluorescence intensity of APHM-13 increased with increasing concentration of human PPARα-LBD in the binding buffer (Figure 11E). However, for human PPARδ-LBD, APHM-13 exhibited the opposite effect (Figure 11F).



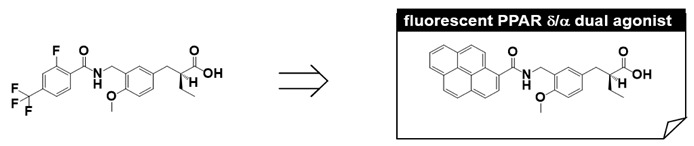



These bi-directional fluorescence properties, depending on complexation with either human PPARα-LBD or human PPARδ-LBD, were worthy of further exploration. Fluorescence intensity is normally expected to increase upon binding to the target protein because of the hydrophobic nature of the target protein binding pocket. In this context, the increase in fluorescence intensity of APHM-13 upon binding to human PPARα-LBD is reasonable. There must also be an explanation for the concentration-dependent attenuation of APHM-13 fluorescence upon binding to human PPARδ-LBD. Figure 11G,H show the binding modes of APHM-13 complexed with human PPARα-LBD and TIPP-401 complexed with human PPARδ-LBD, respectively. The pyrene ring of APHM-13 is buried in the Y2 arm of the human PPARα LBD, which is composed of the side chains of the various hydrophobic amino acids. It is noteworthy that all these amino acids are aliphatic. However, for the human PPARδ-LBD, the Y2 arm is composed of side chains of Ile249, Leu255, Glu259, Trp264, Val281, Arg284, Val341, Val348, Phe352 and Leu353, i.e., the Y2 arm of the human PPARδ-LBD contains two aromatic amino acids, Trp264 and Phe352. The trifluoromethyl group of TIPP-401 is less than 2 Å away from Trp264. Therefore, Trp264 might interact with the pyrene moiety of APHM-13 when APHM-13 is bound to the human PPARδ-LBD.

The naturally occurring aromatic amino acids, tryptophan, tyrosine and phenylalanine, are all intrinsically fluorescent, but tryptophan is the most powerful fluorophore of the three [51]. We speculate that fluorescence quenching involving Trp264 might occur. When APHM-13 in the complex with human PPARδ-LBD is excited, the obtained fluorescence energy of the excited-state pyrene might be transferred to the nearby Trp264 and attenuated.

To confirm the validity of our hypothesis, we constructed four human PPARs-LBD mutants, i.e., 264Trp of the human PPARδ-LBD was changed to Leu (hPPARδW264L-LBD) and the corresponding Leu of the human PPARα-LBD was changed to Trp (hPPARαL258W-LBD), 264Trp of the human PPARδ-LBD was changed to Ala (hPPARδW264A-LBD), and 256Trp of the human PPARδ-LBD was changed to Ala (hPPARδW256A-LBD). We examined the effects of these mutated human PPAR-LBDs on the fluorescence properties of APHM-13. 264Trp but not 256Trp is required for the change of fluorescence properties of APHM-13. The fluorescence of APHM-13 was augmented in the presence of increasing amounts of hPPARδW264L-LBD or hPPARδW264A-LBD, but not of hPPARδW256A-LBD. The fluorescence of APHM-13 was decreased in the presence of increasing amounts of hPPARδW225A-LBD (Figure 12B). These results indicated that the critical amino acid associated with the bi-directional fluorescence properties of APHM-13 is residue 264 located in the omega loop position of human PPARs-LBD.

Overall, we created dual-format human PPARδ and human PPARα ligand binding assays simply by exploiting fluorometric changes elicited by the binding of endogenous and/or exogenous ligands.

## 10. PPAR Pan Agonist: TIPP-703

We have shown that the structure of 3,4-disubstituted phenyl propanoic acid is a good PPAR pharmacophore, especially for PPARα and PPARδ. However, we consider that this framework is not suitable for PPARγ. We therefore aimed to create a 3,4-disubstituted phenyl propanoic acid PPAR agonist, preferentially with PPARγ agonistic activity. To achieve this, we reassessed the three-dimensional structural differences among PPAR LBDs, especially the hydrophobic binding pocket hosting the hydrophobic tail of our PPAR agonists. We noticed that the two pairs of amino acids that interpose the substituent attached at the 4-position of the distal benzene ring of our agonists were different. These were Arg and Trp for PPARδ, Cys and Leu for PPARα, and Gly and Leu for PPARγ (Figure 13A). We believed that the PPARγ LBD possesses the widest binding pocket for hosting the 4-position of the substituent. Therefore, we designed and synthesized compounds with a bulky substituent at the 4-position (Figure 13B) and found an adamantyl group (TIPP-703) to be preferable (Figure 13C).



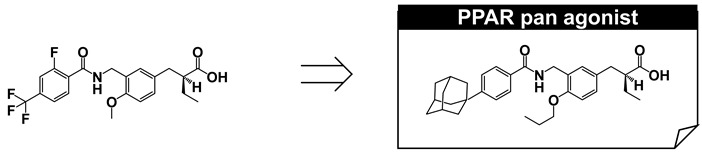



TIPP-703 is a PPAR pan agonist (EC_50_ values of all three PPAR subtypes are approximately 100 nM or less in our assay system) that can simultaneously activate all PPAR subtypes. Considering the beneficial pharmacological effects of each PPAR subtype, a PPAR pan agonist is extremely attractive, especially for the treatment of metabolic syndrome, stroke, heart failure, sudden cardiac death, and certain cancers [52]. Simultaneous activation of all PPARs might also reduce the occurrence of adverse side effects, such as weight gain, fluid accumulation, and pulmonary and macular edemas, which are often associated with PPARγ agonists, such as rosiglitazone and pioglitazone [7].

We compared the ability of MCC-555 [53], rosiglitazone (PPARγ agonists), and TIPP-703 to stimulate adipocyte differentiation of murine preadipocyte 3T3-L1 cells (Figure 14B). PPARγ agonists promote the conversion of a variety of preadipocyte and stem cell lines into mature adipocytes [54]. All three compounds dose-dependently stimulated adipocyte differentiation and induced triglyceride accumulation. The ability of TIPP-703 to differentiate preadipocytes is similar to that of rosiglitazone, and far more potent than that of MCC-555.

To further characterize TIPP-703 as a PPAR pan agonist, we assessed its effects on representative genes having a PPRE in the promoter region in human hepatocellular carcinoma Huh-7 cells (Figure 14A). Carnitine palmitoyl acyl-CoA transferase 1A (*CPT1A*), HMG-CoA synthase 2 (*HMGCS2*), adipocyte differentiation-related protein (*ADRP*) and angiopoietin-like protein 4 (*ANGPTL4*) were selected, as human genes that possess a PPRE in the promoter region [55,56].

Treatment with TIPP-703 augmented the expression of all four genes, to a similar extent to that obtained with the positive control agents. These results indicate that TIPP-703 is an effective PPAR pan agonist and can activate various kinds of PPAR-regulated genes at the cellular level. We also evaluated the effect of TIPP-703 as an anti-pancreatic cancer agent. Expression of PPARγ was confirmed in two pancreatic cell lines, PANC-1 and PT-45 (Figure 15A). When PT-45 cells were treated with 1 μM TIPP-703, induction of G1 phase arrest was observed (Figure 15B). Biochemical studies indicated that TIPP-703 treatment enhanced expression of the cyclin-dependent kinase inhibitor, p21 (Cip1/Waf1) [57], and caused subsequent attenuation of cyclin D1 expression in both PANC-1 and PT-45 cells. p27 expression was not affected by TIPP-703 treatment (Figure 15C) [58]. These results provide strong evidence that activation of PPARs is a promising mechanism for the treatment of pancreatic cancer. Further in vivo experiments are ongoing.

## 11. PPARγ-Selective Agonist: MO-4R

We next aimed to improve PPARγ selectivity and to create a PPARγ-selective agonist using a phenyl propanoic acid framework. We reviewed the three-dimensional structural differences among PPAR LBDs, focusing on the binding pocket hosting the side chain alkyl group of the acidic propanoic acid moiety. We noticed the difference in the two aromatic amino acids in the PPARγ and PPARα LBDs, which are positioned at the pocket entrance of the side chain alkyl group of the propanoic acid moiety. For PPARγ this was His and for PPARα the bulkier Tyr (Figure 16A). We also focused the different aliphatic amino acids in PPARγ and PPARδ LBDs, positioned at the bottom of the cavity hosting the side chain alkyl group. For PPARγ this was Leu and for PPARδ the longer Met (Figure 16A). We imagined that the binding pocket of the PPARγ LBD hosting the side chain alkyl group of our propanoic acid moiety was narrower and deeper. To match the situation, we designed and synthesized compounds with a benzyl group and the related substituent as the side chain alkyl group of the α-position of the phenylpropanoic acid moiety (Figure 16B). We found that a benzyl group was preferable for a PPARγ-selective agonist.



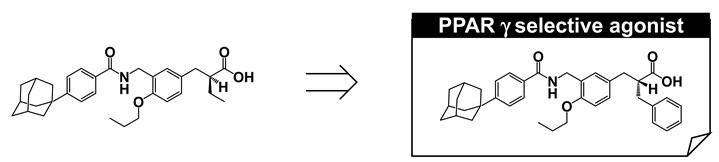



We identified a proper substituent at the α-position for PPARγ-selective activity. However, to our surprise, the stereo-selectivity of the α-benzyl phenyl propanoic acid agonist activity for PPARγ was different. The (*R*)-enantiomer (MO-4R) was a more potent PPARγ-selective agonist than the antipodal (*S*)-enantiomer (MO-3S). The EC_50_ values of MO-4R and MO-3S transactivation activities for PPARγ were 3.60 and 22.0 nM, respectively. Consistently, MO-4R was more potent at inducing 3T3-L1 adipocyte differentiation than MO-3S (Figure 17A).

The structure–activity relationship from our PPAR agonist study indicated that the (*S*)-enantiomer was more potent than the (*R*)-enantiomer. To understand the preference of the benzyl group and the reversal of stereo-selectivity, we performed X-ray crystallographic analysis of both MO-4R and MO-3S complexed with the human PPARγ LBD, and compared the results with those obtained from TIPP-703. As mentioned above, the α-ethylphenylpropanoic acid derivative, TIPP-703, transactivated all PPAR subtypes, while the α-benzylphenylpropanoic acid derivatives, MO-3S and MO-4R, specifically transactivated PPARγ more potently than TIPP-703. These differences in selectivity and activity can be attributed to differences in the interaction potential of the phenylpropanoic acid side chain α-position with the surrounding amino acids. As expected, the volume of the α-substituent-binding pocket of human PPARδ was smaller than that of human PPARα because of the bulky Met residue (Val in human PPARα) located at the bottom of the pocket. The bulky benzyl side chain of MO-3S might not fit into the pocket of human PPARδ because of steric interference with the Met residue. Additionally, the width of the entrance of the human PPARα pocket is smaller than that of human PPARδ because of the bulky Tyr residue (His in human PPARδ) located at the entrance of the pocket. The bulky benzyl side chain of MO-3S might not be able to enter the pocket of human PPARα due to steric hindrance. These results explain why PPARγ agonist activity is improved by exchanging the side chain at the α-position of phenylpropanoic acids. For TIPP-703, the ethyl group interacts with the hydrophobic cavity composed of Phe282, Cys285, Ser289, Tyr327 and His449. however, for MO-3S, the benzyl group is wider than the ethyl group and, therefore, the benzyl group interacts further with the additional hydrophobic cavity composed of Gln286, His323, Phe363, Leu453 and Tyr473. This expansion of the interaction space might be the primary reason for the increase in MO-3S activity compared with that of TIPP-703 (Figure 16C).

Contrary to our expectations, the three-dimensional structures of bound MO-3S and MO-4R closely resemble each other (Figure 17B). The only difference is in the direction of the methylene chain and the configuration of the benzene ring of the α-substituent benzyl group. Surprisingly, although the stereochemistry is opposite between MO-3S and MO-4R, the benzyl side chain of both ligands is located in the same hydrophobic pocket of the human PPARγ LBD formed by H3, H5, H7, H11 and H12 (Figure 17B). We focused on Ser289 and Leu469 in the human PPARγ LBD complexed with MO-3S or MO-4R because Ser289 is the closest amino acid to the methylene chain and Leu469 is the closest amino acid to the distal benzene ring of the side-chain benzyl group of both MO-3S and MO-4R. A short contact between the 2-position hydrogen atom of the MO-3S benzyl group and the oxygen atom of the Ser289 side chain was observed, together with a short contact between the 3-position hydrogen atom of the MO-3S benzyl group and the hydrogen atom of the Leu469 side chain. However, such short contacts were not observed for MO-4R. This may be one reason why MO-3S is a weaker human PPARγ agonist than MO-4R (Figure 17C).

## 12. PPARγ-Partial Agonist: MEKT-21

We have created MO-4R, a structurally new and very potent PPARγ-selective agonist, that is distinct from the well-known TZD class PPARγ-selective agonists. We therefore started an anti-cancer drug-development project for MO-4R. We performed various evaluations; however, to our disappointment, we were unable to prepare sufficiently high concentrations of MO-4R in aqueous solution for toxicological studies. We temporally suspended our project and turned our attention to improvement of the aqueous solubility of MO-4R. Aqueous solubility of compounds is an important issue in medicinal chemistry research, especially during structural development and/or structural optimization because this parameter must be optimized to ensure the drug possesses suitable in vivo pharmacological activity.



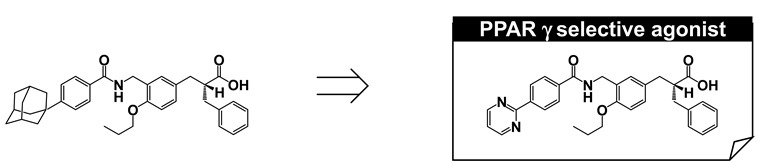



To create soluble phenylpropanoic acid-type PPARγ-selective agonists, we re-focused on the substituent at the hydrophobic tail. The poor aqueous solubility of MO-4R might be primarily attributed to the presence of an adamantyl group in the hydrophobic tail, even though this substituent is the critical determinant for potent human PPARγ-selective agonism. We have successfully created biphenylcarboxylic acid-type human PPARδ agonists with improved aqueous solubility [59]. This improved solubility was attributed to two factors, disruption of molecular planarity and decreased hydrophobicity via introduction of a heteroaromatic ring. We applied the same strategies to the present work and found that a pyrimidine-2-yl group (MEKT-21) was preferable (Figure 18A). MEKT-21 exhibited an approximately 10,000-fold increased aqueous solubility, as judged from the calculated log(partition coefficient). In accordance with this, MEKT-21 was eluted more quickly than MO-4R. MO-4R exhibited extremely poor aqueous solubility (<1 μg/mL), but MEKT-21 improved the solubility to 301 μg/mL. We selected MEKT-21 for further evaluation, based on its activity towards human PPARγ and good aqueous solubility.

It is interesting to note that evaluation of intrinsic MEKT-21 activity indicated that it is a partial agonist, not a full agonist; PPARγ was activated to approximately 65% of the maximum activity achieved with the full-agonist, pioglitazone (Figure 18B).

To investigate the potential of the PPARγ partial agonist MEKT-21 as an anti-scirrhous gastric cancer agent, we compared the apoptosis-inducing effects of MEKT-21 and MO-4R, together with the human PPARγ full agonist, troglitazone, on the human scirrhous gastric cancer cell line OCUM-2MD3 [60]. Troglitazone was used as a positive control because it suppresses the growth of gastric cancer cell lines, such as MKV45 [61], SNU-216 and SNU-668 cells [62]. Troglitazone dose-dependently enhanced apoptosis in the concentration range 1–100 μM. Therefore, troglitazone induced apoptosis of OCUM-2MD3 cells but had little effect on normal cells. The human PPARγ full agonist, MO-4R, also dose-dependently enhanced apoptosis in the concentration range 1–100 μM, and it was more potent than troglitazone (10 μM MO-4R effectively induced apoptosis of OCUM-2MD3 cells). However, 100 μM MO-4R induced apoptosis of both OCUM-2MD3 and OCUM-24 cell lines. Therefore, MO-4R has poorer selectivity for human scirrhous gastric cancer cells compared with troglitazone (Figure 18C).

The human PPARγ partial agonist, MEKT-21, also dose-dependently induced apoptosis of OCUM-2MD3 cells in the concentration range 1–100 μM. Its potency was equal to or somewhat greater than that of troglitazone (based on activity at 10 μM). Furthermore, MEKT-21 showed high selectivity for cancer cells, comparable to that of MO-4R. These results indicated that partial activation of PPARγ might be sufficient to induce apoptosis of human scirrhous gastric cancer cells.

X-ray crystallography (Figure 19) clearly revealed the reason for this difference. The pyrimidin-2-yl ring and adamantyl ring are hosted in the same binding pocket, i.e., the large Y3 arm cavity composed of the side chains of amino acid residues Leu255, Glu259, Ile262, Phe264, Arg280 and Ile281. The planar pyrimidin-2-yl ring can interact less effectively with this large hydrophobic cavity compared with the three-dimensionally expanded hydrophobic adamantyl ring, especially with regard to hydrophobic interactions with Ile262, Phe264, and Ile 281, which are positioned on both sides of the pyrimidin-2-yl ring of MEKT-21 (Figure 19D–G).

The qualitative characteristics of corepressor dissociation and coactivator recruitment of nuclear receptors are regulated differentially by the three-dimensional structure around the C-terminal H12 and the surrounding helixes [63]. In the binding of a full agonist, such as rosiglitazone, H12 forms part of the coactivator-binding surface, along with H3 and H5 [64]. In this case, the positions and directions of the side chains of two distinct hydrophilic amino acids, Lys301 (in H3) and Glu471 (in H12), are critically important because these amino acids clamp the LXXLL motif (L, leucine; X, any amino acid) of the coactivator, and dock it appropriately to express full-agonistic activity. Therefore, we compared the three-dimensional structures of two PPARγ full agonists, rosiglitazone and farglitazar [65], complexed with the PPARγ LBD (Figure 20A–D). The overall folds are well matched and the positions and the directions of the side chains of Lys301 and Glu471 overlap well in these structures. The root mean square deviations (RSMDs) of these two amino acid residues are 0.71 Å and 0.44 Å, respectively (Figure 20E,F). These results indicate that the two PPARγ full agonists can recruit coactivator complexes with almost the same efficacy, eliciting full agonistic activity.

However, the overall structures of the PPARγ LBD complexed with PPARγ partial agonists, including MEKT-21, are somewhat different. The frameworks of the PPARγ LBD complexed with several representative PPARγ partial agonists (MEKT-21, LT127 [66], phenoxyacetic acid compound [67], GW-0072 [68], pyrazole compound [69], and GQ-16 [70]) are depicted in Figure 21A–F, and in a superposition in Figure 21G. The folds of the upper half of the PPARγ LBD are well matched in these complexes. However, the folds of the lower half of the PPARγ LBD are somewhat different. The position of Lys301 in H3 varies somewhat in these structures (RMSD = 1.10 Å compared with 0.71 Å for full agonists). However, the side chain of Glu471 in H12 faces in various directions in these complexes, and the RSMD is much larger (1.89 Å compared with 0.44 Å for full agonists). These substantial differences in structure might be one of the driving forces behind the partial agonistic activity of these compounds, including MEKT-21.

To further understand the key differences in the binding of full and partial PPARγ agonists, we focused on the binding mode of these agonists to Tyr473 in H12. Tyr473 is believed to play a critical role in the full agonist-mediated stabilization of the LBD of PPARγ (Figure 22A–C). For rosiglitazone, the side chain phenol oxygen of Tyr473 forms a tight hydrogen bond with the nitrogen atom of the thiazolidine-2,4-dione ring. Furthermore, the side chain phenol oxygen of Tyr473 has a hydrogen bond interaction with the carbonyl oxygen atom of the thiazolidine-2,4-dione ring. These hydrogen bond networks tightly fix Tyr473 in H12, forming a section of the transcriptional coactivator-binding surface (Figure 22A).

In the partial agonist, *N*-[1-(4-fluorophenyl)-3-(2-thenyl)-1*H*-pyrazole-5-yl]-3,5-bis-(trifluoromethyl)benzenesulfonamide [Protein Data Bank (PDB): 2GOH], the 4-fluorophenyl group is located close to H12, but it has no hydrophobic or hydrogen bond interaction with this helix and there is no significant ligand-mediated interaction to fix Tyr473 in H12. The absence of interaction with Tyr473 in H12 might provide the structural basis for the partial agonist functionality of not only this compound, but also the other neutral-and carboxylic acid-type PPARγ partial agonists (Figure 22C).

The acidic carboxylic acid moiety of α-benzylphenylpropanoic acid MEKT-21 hydrogen bonds with the side chain phenol oxygen of Tyr473. However, the distance between these two functionalities is longer than that of the PPARγ LBD complexed with the thiazolidine-2,4-dione-type PPARγ full agonist, rosiglitazone. Furthermore, there is only a single hydrogen bond between the acidic moiety of MEKT-21 and Tyr473 in H12. Tyr473 in H12 is thought to be fixed less effectively when MEKT-21 is bound to the PPARγ-LBD, resulting in partial agonistic activity (Figure 22B).

## 13. Structural Basis of Full and Partial PPARγ Agonists

TZD class full PPAR*γ* agonists are widely used for the treatment of type 2 diabetes. However, TZDs possess adverse side effects, including significant weight gain, peripheral edema, bone loss and increased risk of congestive heart failure, which are associated with over-activation of PPARγ [71]. Recently, attention has turned to partial PPARγ agonists, which activate PPARγ less than maximally.

We created the partial PPARγ agonist, MEKT-21. The exact mechanism by which MEKT-21 exhibits partial PPARγ agonism is not fully understood; however, based on clinical need, we aimed to design and synthesize further partial PPARγ agonists. Based on the NR transactivation mechanism, an H12 holding inducer is a full NR agonist. We noted that a tight hydrogen bond network is critical to induce proper H12 holding for maximum activity of PPARγ agonists. Therefore, we predicted that replacement of the PPARγ agonist carboxyl group with another weak acidic functionality, such as acylsulfonamide, might result in partial PPARγ agonists [72]. An acylsulfonamide group interacted with Tyr473 moderately to partially stabilize H12 of the PPARγ LBD. We modified the structure of the full agonist, TIPP-703, to remove the carboxyl group and to introduce an acylsulfonamide group to obtain MEKT-75 (Figure 23A).

As expected, MEKT-75 exhibited apparent transactivation activity, although without clear dose-dependency, and its maximal activity was approximately 50% of the maximal response of the positive controls (Figure 23B). These results are consistent with the idea that MEKT-75 is a partial PPARγ agonist.

We solved the X-ray crystallographic structure of MEKT-75 complexed with the PPARγ LBD homodimer (Figure 24A–L). The crystal was obtained by soaking a crystal of the homodimer in ligand solution. Each PPARγ LBD in the homodimer binds one molecule of MEKT-75. Interestingly, the structural folding of the LBD remains almost unchanged, except for the region from the end of H11 to the C-terminal H12 region. Furthermore, the structural folds in one part of the homodimer structure are almost identical to those in the PPARγ LBD–rosiglitazone complex (Figure 24B), whereas those in the other part of the homodimer structure are different (Figure 24C). Therefore, we tentatively designated the former structure as the fully active form of the PPARγ LBD, and the latter structure as a non-fully active form.

In the fully active form, MEKT-75 takes a U-shaped structure, similar to that of TIPP-703. The acylsulfonamide group of MEKT-75 is positioned near H11. A hydrogen bond network involving five amino acids, Ser289, His323, Tyr327, His449 and Tyr473, is formed in the interaction of full agonists with the LDB. Therefore, interactions of three of these five amino acids are conserved in the one LDB complexed with MEKT-75. This appears to be sufficient to support the fully active LDB structure [Figure 24K,L (left)]. In contrast, in the non-fully active LDB, another interaction was noted: His266 is located close to the (pyrimidin-2-yl)phenyl moiety of MEKT-75, resulting in hydrophobic interaction between His266 and the phenyl group, and between the His266 and the pyrimidin-2-yl group. These additional interactions cause the bound MEKT-75 to move to the right, so that the distance from the phenylsulfonylaminocarbonyl group of MEKT-75 to the side chains of Ser289, Tyr327, Lys367, His449 and Phe363 becomes longer. As a result, the hydrogen bond network is weaker, which may mean that the H12 region is not restricted to the appropriate location for full activity [Figure 24K,L (right)].

We checked the X-ray crystallographic structure of the PPARγ LBD apo-form (without ligand). The PPARγ LBD apo-form also forms a homodimer in which one LDB is in the fully active form, and the other is in a non-fully active form [Figure 26A (center)]. This indicates that two types of PPARγ LBD structures are present in the crystal state, irrespective of the presence or absence of agonist. Indeed, the apo-form PPARγ LBD homodimer and the ligand-bound PPARγ LBD homodimer did not show apparent structural differences in the LDB, indicating that partial agonists lack the ability to induce a fully active LBD [Figure 26A (left)].

In contrast, the PPARγ LBD-rosiglitazone complex also formed a homodimer in the crystal, but each LBD was present in a fully active form [Figure 25A and Figure 26A (right)]. This result indicates that full agonists do induce structural change of the non-fully active PPARγ LBD to a fully active LBD, presumably by facilitating a tight hydrogen bond network with the LBD. We speculate that there is a dynamic equilibrium between the fully active and non-fully form of the PPARγ LBD.

To ascertain the generality of our observations, we surveyed the PDB database. Representative PPARγ LBD homodimer–partial agonist structures are depicted in (Figure 25B–D: PDB: 2I4Z [73], PDB: 2Q5S [66], and PDB: 2Q6R [74]). For PDB: 2I4Z, the homodimer contains a fully active and a non-fully active LBD, and the ligand is bound to the fully active LBD. For PDB: 2Q5S, the homodimer also contains the two LBD forms, and the ligands are bound to both LBDs. For PDB: 2Q6R, the homodimer contains two non-fully active LBDs, and the ligands are bound to both LBDs. Based on these data, we suggest that partial PPARγ agonists lack the ability to induce a fully active LBD.

One of the current trends of NR agonist discovery is focused on partial NR agonists, which activate NR function less than maximally to reduce adverse side effects elicited by full NR agonists. The structural biology study described here, based on a partial H12 holding induction design, might be one option to realize such drugs.

## 14. Concluding Remarks and Future Directions

In this review, we have described our more than 20 years of studies designing and synthesizing PPAR agonists based on the 3,4-disubstituted phenylpropanoic acid structure as a versatile template for subtype-selective PPAR agonists. Our working hypotheses for design involve the ligand superfamily concept and the H12 holding induction/partial induction concept, which enable the creation of various PPAR-subtype-selective agonists and partial agonists. The structure–activity relationships among agonists for PPARs were well characterized and are summarized in Figure 27.

We think the present strategy is applicable to the development of other nuclear receptor agonists/partial agonists because the structural similarity of LBDs in the various nuclear receptors is high. In support of this, we have succeeded in creating vitamin D receptor agonists [75,76], and farnesoid X receptor agonists [77,78].

Furthermore, considering the ligand-mediated activation mechanism of action, the present strategy is also applicable to NR antagonist design. Accordingly, we have also been engaged in the design and synthesis of NR antagonists for PPARδ [79], PPARγ [80], LXR [81,82], farnesoid X receptor [78], and progesterone receptor [83]. These results will be disclosed in the near future.

## Data Availability

Data sharing not applicable.

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
