# Peer review of "Structural Biology-Based Exploration of Subtype-Selective Agonists for Peroxisome Proliferator-Activated Receptors"

_ijms, 2021, doi:10.3390/ijms22179223_

Round 1

Reviewer 1 Report

This is an interesting and well-conceived and clearly written review on an important topic since peroxisome proliferator-activated receptors are involved in the metabolism. Therefore, this is a welcome summary of peroxisome proliferator-activated receptors structural biology based on the author's works which spent 20 years studying that.

Author Response

Reviewer 1 comment

Comments and Suggestions for Authors

C: This is an interesting and well-conceived and clearly written review on an important topic since peroxisome proliferator-activated receptors are involved in the metabolism. Therefore, this is a welcome summary of peroxisome proliferator-activated receptors structural biology based on the author's works which spent 20 years studying that.

  • Thank you very much Reviewe1 for extremely positive comment. The author think this manuscript quite fruitful to all NR-receptor medicinal chemist.

Hiroyuki Miyachi

Reviewer 2 Report

Miyachi summarizes their medicinal chemistry research on the design, synthesis, and pharmacological evaluation of subtype-selective PPAR agonists. The review is extremely detailed and based on their own experience.

Some small modifications are required:

Line 49: Statement not true.

Line 54ff: Very much simplified: extensive description of bindings sites and target genes can be found in PMID: 20026355.

“3. Pleiotropic effect of PPARs” has to be re-written completely considering the progress of the last 20 years.

Scheme 1 should be numbered as a Figure.

Figure 17 C: spelling error.

Figure 20 C: The box is red instead of green. Same color problem in Figure 21.

The abbreviation list should be in alphabetical order.

“The author thanks all collaborators…” should be replaced. If there are certain persons to acknowledge, they must be named.

Reference list: please check carefully for unjustified underlining, hyperlinks, etc.

The reference list is a bit outdated. Please include more recent references regarding PPAR function. There are several excellent reviews for instance from the Wagner or Wahli groups.

Please check carefully to respect the template, i.e. unify fonts and size.

Author Response

Reviewer 2 comment

Comments and Suggestions for Authors

C: Miyachi summarizes their medicinal chemistry research on the design, synthesis, and pharmacological evaluation of subtype-selective PPAR agonists. The review is extremely detailed and based on their own experience.

è Thank you Reviewer2 for positive consideration. The author think this manuscript very fruitful to all medicinal
chemists.

Some small modifications are required:

Q1: Line 49: Statement not true.

è Thank you Reviewer2 for point out this. According to Reviewer2’s comment, the author revised as, “PPARδ is expressed in adipose tissue, skeletal muscle, heart and so on [3].” 

Q2: Line 54ff: Very much simplified: extensive description of bindings sites and target genes can be found in PMID:
20026355.

è The author agrees with the comment. The author described the essencial.

Q3: “3. Pleiotropic effect of PPARs” has to be re-written completely considering the progress of the last 20 years.

è Thank you Reviewer2 for point out this. According to Reviewer2’s comment, the author added the sentence, “Recent extensive biological studies clearly disclosed PPARs function beyond metabolism. Each PPAR subtypes play major roles in a broad spectrum of biological processes, including cell proliferation and differentiation, fatty acid and eicosanoid signaling, bone formation, tissue repair and remodeling, insulin sensitivity [12].” The author add new Ref.12.

Q4: Scheme 1 should be numbered as a Figure.

è The author think this a chemical reaction scheme, so the author use “Scheme” and would like to remain as it is.

Q5: Figure 17 C: spelling error.

è Thank you Reviewer2 for point out this. According to Reviewer2’s comment, the author revised properly.

Q6: Figure 20 C: The box is red instead of green. Same color problem in Figure 21.

è Thank you Reviewer2 for point out this. According to Reviewer2’s comment, the author revised properly.

Q7: The abbreviation list should be in alphabetical order.

è Thank you Reviewer2 for point out this. According to Reviewer2’s comment, the author revised properly.

Q8: “The author thanks all collaborators…” should be replaced. If there are certain persons to acknowledge,
they must be named.

è Thank you Reviewer2 for point out this. According to Reviewer2’s comment, the author picked up two important
contributors in the revised manuscript.

Q9: Reference list: please check carefully for unjustified underlining, hyperlinks, etc.

è Thank you Reviewer2 for point out this. According to Reviewer2’s comment, the author revised properly.

Q10: The reference list is a bit outdated. Please include more recent references regarding PPAR function.
There are several excellent reviews for instance from the Wagner or Wahli groups.

è Thank you Reviewer2 for point out this. According to Reviewer2’s comment, the author revised some refs.

Q11: Please check carefully to respect the template, i.e. unify fonts and size.

è Thank you Reviewer2 for point out this. According to Reviewer2’s comment, the author revised
 (this might be some technical errors occured).

Hiroyuki Miyachi